# The Role of CXC Chemokines in Cancer Progression

**DOI:** 10.3390/cancers15010167

**Published:** 2022-12-28

**Authors:** Tiantian Wu, Wannian Yang, Aiqin Sun, Zhixiao Wei, Qiong Lin

**Affiliations:** School of Medicine, Jiangsu University, Zhenjiang 212013, China

**Keywords:** CXC chemokines, CXCR, angiogenesis, inflammation, metastasis, therapy

## Abstract

**Simple Summary:**

CXC chemokines are small molecules and secretory peptides, which are significantly associated with cancer progression. Aberrant expression of CXC chemokines is always observed in cancer patients. An increasing number of studies have reported that CXC chemokines play essential roles in tumor angiogenesis, tumor-promoting inflammation, and metastasis, which favor cancer promotion. The expression level of CXC chemokines is closely associated with the clinicopathological characteristics and outcomes of cancer. The specific roles of CXC chemokines and corresponding signaling pathways in cancer progression are summarized in this review. Furthermore, we have discussed the potential application of CXC chemokines for cancer targeted therapy and personalized treatment.

**Abstract:**

CXC chemokines are small chemotactic and secreted cytokines. Studies have shown that CXC chemokines are dysregulated in multiple types of cancer and are closely correlated with tumor progression. The CXC chemokine family has a dual function in tumor development, either tumor-promoting or tumor-suppressive depending on the context of cellular signaling. Recent evidence highlights the pro-tumorigenic properties of CXC chemokines in most human cancers. CXC chemokines were found to play pivotal roles in promoting angiogenesis, stimulating inflammatory responses, and facilitating tumor metastases. Enhanced expression of CXC chemokines is always signatured with inferior survival and prognosis. The levels of CXC chemokines in cancer patients are in dynamic change according to the tumor contexts (e.g., chemotherapy resistance and tumor recurrence after surgery). Thus, CXC chemokines have great potential to be used as diagnostic and prognostic biomarkers and therapeutic targets. Currently, the molecular mechanisms underlying the effect of CXC chemokines on tumor inflammation and metastasis remain unclear and application of antagonists and neutralizing antibodies of CXC chemokines signaling for cancer therapy is still not fully established. This article will review the roles of CXC chemokines in promoting tumorigenesis and progression and address the future research directions of CXC chemokines for cancer treatment.

## 1. Introduction

According to global cancer statistics, there were 19.3 million new cancer cases and almost 10 million cancer deaths in 2020 around the world [1]. New cases of cancer around the world have been predicted to reach 28.4 million by 2040. Thus, investigation of the molecular mechanisms underlying tumor genesis and progression and identification of new therapeutic targets are necessary for developing more effective cancer therapeutic drugs and techniques. Previous studies have shown that chemokines (e.g., CXC chemokines and CC chemokines) play vital roles in the crosstalk between tumor cells and their microenvironments. It has been demonstrated that chemokines are highly expressed in many cancer types, and their expression is positively associated with cancer angiogenesis, inflammation, metastasis, and poor survival profiles. Chemokines are low molecular weight (8–10 kDa) peptides that are inducible and predominantly chemotactic for leukocytes [2]. Chemokines are classified based on amino acid numbers between the first and the second cysteine residues in the peptide sequence and are divided into four subfamilies: (a) C, (b) CC, (c) CXC, and (d) CXXXC, where X stands for any amino acid residues [3]. Among them, CXC and CC subtypes are two major subgroups of the chemokine family [4]. A growing number of studies have shown that altered expression of these chemokines is closely associated with tumor progression. CCL2, one of CCL-type inflammatory cytokines, was observed to promote cancer progression and metastasis in breast, ovarian, and prostate cancer through recruiting pro-tumoral (M2) macrophages [5]. In addition, CCL2 was found to cooperate with CXCL6 to chemoattract neutrophils, accompanied by inflammatory cells infiltration in tumor sites [6]. CXC chemokines play roles in tumorigenesis through autocrine and paracrine pathways. Besides cancer cell-secreted chemokines, CXC chemokines are also derived from many other kinds of cells (e.g., tumor-associated macrophages [7,8,9], neutrophils [10], endothelial cells [11,12], cancer-associated fibroblasts [13,14], mesenchymal stem cells [15], and stromal cells [16,17] ) and then empower various types of cancer with more malignant phenotypes in a paracrine fashion. CXC chemokines could be induced by other cytokines to promote tumor progression. A study reported that fibroblast growth factor-2 (FGF-2), secreted by cancer-associated fibroblasts, stimulates CXCL8 release and results in pancreatic tumor progression [18]. Enforced expression of CC chemokines, CXC chemokines, and many other cytokines favors premalignant diseases developing into aggressive cancers. Therefore, figuring out the relationship between these factors and tumor progression is a prerequisite for improvement of survival outcomes in cancer patients. To date, great efforts have been made to understand the roles of the CXC chemokines in human cancers. This review will focus on discussing the effects of CXC chemokines on tumor angiogenesis, inflammation, and metastasis and will address potential therapeutical applications of CXC chemokines for cancer treatment.

## 2. CXC Chemokine Family and the CXCL/CXCR Signaling Axes

CXC chemokines are small secretory proteins with four highly conserved cysteine amino acid residues and the first two cysteines are separated by one non-conserved amino acid residue [19]. Based on the presence or absence of a Glu-Leu-Arg (ELR) motif in the NH2-terminus, CXC chemokine family members are divided into two subtypes: ELR+ members and ELR- members [20]. The ELR+ members include CXCL1, CXCL2, CXCL3, CXCL5, CXCL6, CXCL7, CXCL8, and CXCL17, while the ELR- members contain CXCL4, CXCL9, CXCL10, CXCL11, CXCL12, CXCL13, CXCL14, and CXCL16 [21,22,23]. 

The CXC chemokines (e.g., CXCL1, CXCL2, CXCL3, CXCL5, and CXCL8) are significantly upregulated in most cancers and positively associated with cancer metastasis and chemo-resistance. Conversely, downregulation of these CXC chemokines greatly suppresses the motility of cancer cells [24]. Altogether, CXC chemokines play the following roles in tumor progression: (1) to regulate angiogenesis, either angiogenic or angiostatic; (2) to recruit multiple types of leukocytes and mediate inflammation process; and (3) to promote tumor metastases. Because angiogenesis, inflammation, and metastasis are main hallmarks of cancer [25], CXC chemokines have great value to be used for cancer diagnosis and prognosis.

CXC chemokine receptors, CXCRs, are G protein-coupled and seven-transmembrane receptors for CXC chemokines [3,26]. The CXCR family contains seven members: CXCR1–CXCR7. The ELR+ CXCLs are the ligands for CXCR1 and/or CXCR2 [21,26], while the ELR- CXCLs are primarily the ligands for CXCR3, CXCR4, CXCR5, CXCR6 or CXCR7. In general, ELR+ CXCLs/CXCR1/2 signaling promotes tumor progression while ELR- CXCLs/CXCR3-7 signaling mainly has tumor-suppressive effects. However, in some cases, some ELR- CXCLs/CXCR3-7 signaling also has pro-tumorigenic properties, such as ELR- CXCL4/CXCR3 [27], ELR- CXCL12/CXCR4 [28,29], ELR- CXCL11/12-CXCR7 [26], and ELR- CXCL13/CXCR5 [30]. In this review, we mainly focus on the pro-tumorigenic effects of CXCLs/CXCRs signaling. The detailed information about the classification, signaling pathways, and physiological and pathological effects of the CXC chemokines is summarized in Figure 1 and Table 1.

Previous studies have shown that the expression of CXCR is not only observed in tumor cells, but also in granulocytes, monocytes, mast cells, some natural killer cells [58], endothelial cells, and myeloid cells [59], promoting tumor growth and vasculature [2]. Several studies found that upregulation of CXCR1 or CXCR2 resulted in an increase of melanoma cell proliferation and invasion, while knockdown of CXCR1 and/or CXCR2 led to inhibition of melanoma cell growth, motility, and vascularization both in vitro and in vivo [60,61]. Moreover, CXCR2 overexpression is linked to inferior prognosis and survival outcomes [62]. CXCR7 is the receptor of CXCL11 and CXCL12 ligands. It has been implicated in angiogenesis [41]. Furthermore, CXCR7 was observed to enhance the production of proangiogenic factors interleukin-8 (IL-8) and vascular endothelial growth factor (VEGF), thus elicit neovascularization in bladder cancers [63]. Additionally, CXCR4 is another receptor for CXCL12 ligand. Studies have shown that CXCR4 could stimulate cancer progression by dominating the RhoA/ROCK pathway [39]. These studies indicate that targeting CXCRs may have great potential for cancer therapy. In a murine model of lung cancer, administration with anti-CXCR2 antibodies dramatically inhibited lung metastases [64]. In addition to neutralizing antibodies, multiple antagonists of CXCRs, such as SB225002, AZD5069, reparixin, danirixin, SB-656933, navarixin, and SX-682, were observed to suppress tumor growth [62,64]. 

The CXCR signaling contexts are determined by CXCLs and/or tumor types. CXCRs are involved in variable signaling pathways which regulate diverse biological activities in tumor cells. CXCR2 is a major receptor for ELR+ CXCLs, including CXCL1, CXCL2, CXCL3, CXCL5, CXCL6 [52], CXCL7, and CXCL8 [32,65]. CXCLs/CXCR2 axes stimulate tumor-stromal communication, leading to pancreatic cancer promotion [66]. The CXCLs/CXCR2 autocrine loop is also concerned with lung tumorigenesis and mediates tumor cell proliferation and epithelial–mesenchymal transition (EMT) by regulating the p38/ERK MAPK pathway [62]. In papillary thyroid carcinoma cells, the CXCL5/CXCR2 axis was found to enhance mesenchymal marker vimentin and snail expression, thus promoting EMT of the tumor cells [35]. In addition, the EMT effect is also observed in the CXCL12/CXCR4 axis [4]. Moreover, the CXCL12/CXCR4/7 axis could also facilitate cancer cells proliferation and survival through ERK, AKT, and Ras signaling pathways [42,43]. In addition, activation of CXCR2 by CXCL1 was observed to facilitate cancer cell migration and invasion in oral squamous carcinoma [13]. Furthermore, the CXCL1/CXCR2 axis was shown to enhance inflammatory signals in oral squamous carcinoma [48]. Similar tumor-promoting effects were also found in the CXCL3/CXCR2 axis. Sun X et al. discovered that activation of the CXCL3/CXCR2 signaling axis facilitates myofibroblasts transition and then enhances collagen III expression which promote pancreatic cancer metastasis [9]. In addition, overexpression of CXCL8 and CXCR1 or CXCR2 led to tumor cell growth and metastasis via activating PI3K/AKT and ERK1/2 MAPK signaling [32,60]. When the CXCL8-CXCR1/2 pathway was blocked by CXCL8-neutralizing antibodies in a mouse model study, tumor growth, angiogenesis, and metastasis were significantly inhibited [67,68]. The CXCL/CXCR signaling axes also trigger hematologic tumor initiation and development. The CXCL13/CXCR5 axis is confirmed to facilitate tumor cell growth and metastasis in prostate cancer [44] and colorectal cancer [45] via MAPK and PI3K/AKT signaling pathways. Consistent with the solid tumor-promoting function [56], the CXCL13/CXCR5 axis also functions in promoting B-cell acute lymphoblastic leukemia (B-ALL) cell migration and proliferation [57]. Downregulation of CXCL13/CXCR5 signaling by either reducing CXCL13 levels or blocking CXCR5 expression significantly reduced the proliferation of B-ALL cells and favored tumor prognosis, while upregulation of CXCL13/CXCR5 signaling with the addition of exogenous CXCL13 increased the growth of leukemia cells [57]. Overall, CXCL/CXCR signaling axes regulate communication between tumor cells and the tumor microenvironment through multiple signaling pathways (Table 2).

CXCL/CXCR signaling axes play critical roles in cancer development and persistency. Upon binding of CXCLs to CXCRs, CXCR signaling is activated that elicits a series of tumor-associated effects, either pro-tumorigenic or tumor-suppressive. Therefore, CXCR antagonists and neutralizing antibodies towards tumor-promoting chemokines or, alternatively, CXCR agonists of tumor-suppressing chemokine receptors and exogeneous recombinant protein of tumor-suppressing chemokine are able to be used for cancer treatment. There have been many studies on determination of the efficacy of CXCR antagonists and neutralizing antibodies towards CXCLs or CXCRs in various types of cancer. For example, the small-molecule antagonists, SCH-527123 and SCH-479833, that target CXCR2/CXCR1 were found to inhibit colon cancer and melanoma by decreasing neovascularization and increasing apoptosis of tumor cells [72,73]. The anti-tumor effect of CXCR2 antagonist SCH-527123 may be produced by impairing CXCR2 signaling. A study also reported that the combination of SCH-527123 and oxaliplatin optimized the treatment effect of oxaliplatin [74]. A small molecular weight antagonist of CXCR3, AMG487, was observed to block tumor development by inhibiting metastasis [75]. Treatment with CXCR7 antagonists in animal models markedly inhibited tumor growth [76]. CXCR4 inhibitors have been demonstrated to block the growth and migration of head and neck tumors [77], primary brain tumors [78], and breast tumors [79]. Furthermore, neutralizing antibody-targeting CXCR2 prevented pancreatic cancer tumorigenesis through inhibiting angiogenesis [80]. Treatment of NSCLC tumor-bearing mice with neutralizing anti-CXCL5 antibodies decreased tumor growth, tumor vascularity, and spontaneous metastases [81]. Therefore, targeting CXCRs or CXCLs by antagonists and neutralizing antibodies is a favorable treatment option for cancer.

## 3. CXC Chemokines and Tumor Angiogenesis

Angiogenesis is regarded as an essential process to maintain the progression of pathological conditions, including cancer and chronic inflammatory diseases [82]. In tumor microenvironments, constitutive neovascularization supports tumor invasion and metastasis [83]. Statistical analysis shows that approximately 90% of cancer-associated deaths are attributed to tumor metastases [2]. Interrupting neovascularization is an important approach for inhibiting tumor metastasis and reducing cancer mortality. Many studies have revealed the tight connection between CXC chemokines and vascularization. CXC chemokines are chemotaxis of endothelial cells [84]. Existing literature has confirmed that augmented expression of CXC chemokines induces endothelial cells proliferation, migration, and recruitment during tumor micro-vessel formation [83,84,85]. 

The ELR+ members of the CXC chemokine family promote tumor angiogenesis [20,21,22,51,54]. CXCL5, as an ELR+ member, is a powerful angiogenesis activator. Multiple studies have observed that CXCL5 is overexpressed in non-small cell lung cancers (NSCLCs) and its expression was correlated with unfavorable prognosis in patients with NSCLC [33,86]. CXCL8 is another active angiogenic factor. Enhanced secretion of CXCL8 promotes recruitment of endothelial cells for microvascular hyperplasia of tumors [87]. When CXCL8 activates CXCR2, neutrophils infiltrate into tumor tissue and synthesize angiogenic factors, such as VEGF, to stimulate vascularization [88]. A recent study confirmed that CXCL5 and CXCL8 facilitate cyclooxygenase-2 (COX-2)-mediated angiogenesis in NSCLC [48]. Treatment with an anti-CXCR2 antibody abrogated COX-2-mediated angiogenesis [50]. In addition, activation of the CXCL1/CXCR2 signaling axis was also observed to promote tumor angiogenesis and progressive growth by activating STAT3 and enhancing VEGF levels [36]. 

CXC chemokines are upregulated in tumorigenesis. In melanoma tumor, the level of CXCL1, CXCL2, and CXCL3 is significantly elevated, accompanied by proangiogenic activity [2]. Furthermore, in triple-negative breast cancer (TNBC), CXCL1, CXCL2, and CXCL8 are all highly expressed, which is beneficial for tumor blood vessel formation. Knockout of CXCL1/2/8 in TNBC tumors abrogated the increasion in blood vessel formation [37]. 

Most ELR- members of CXC chemokines have an anti-angiogenic function, such as CXCL9 and CXCL10 [53]. However, some of them also have angiogenic properties. CXCL4, an ELR- member, is upregulated in colorectal cancer cells and increases micro-vessel densities through activating the IKKβ/NF-κB pathway [27]. CXCL12, another ELR- member, also promotes colon cancer angiogenesis by activation of MAPKAP kinase 2 signaling [38]. It seems that the function of different CXC chemokines for angiogenesis is determined by the tumor microenvironment. CXC chemokine-promoted angiogenesis is mediated by diverse signaling pathways. Taken together, CXC chemokines play essential roles in tumor angiogenesis and have significant advantage in being targets for intervening tumor angiogenesis.

## 4. CXC Chemokines and Tumor-Associated Inflammation

Tumor-associated inflammation is a key feature of cancers. Many studies have demonstrated that inflammation, especially chronic inflammation, promotes tumor development [89]. Most CXC chemokines are inflammatory, except CXCL12/SDF-1 and CXCL13 which are homeostatic [90]. These inflammatory chemokines are not constitutively expressed but are inducible and upregulated by inflammatory stimuli such as LPS, TNF, IL-1, and IFN-γ and they are able to induce expression and synthesis of CXC chemokines [90,91]. However, one stimulus may affect only several specific chemokines (Table 3).

When the tumor microenvironment is similar to chronic inflammation, the tumor tissues are surrounded with an increased number of neutrophils and deteriorate in some cases [92]. Tumor-associated macrophages are also critical components in the tumor inflammatory microenvironment and induce a more aggressive phenotype of pre-malignant cells [93]. Tumor-associated macrophages are able to inhibit tumor-specific T cells and fight against tumor immunity [94]. In tumor inflammation, CXC chemokines are strong chemoattractants for neutrophils, macrophages, and lymphocytes and are inflammation stimulators [22]. It was observed that CXCL5 mediates inflammation by activating the PI3K/AKT and the MAPK/ERK1/2 signaling pathways to promote cancer progression [33]. In addition, binding of CXCL5 to CXCR2 allows for recruitment and activation of the neutrophils that are involved in inflammatory responses [86]. Moreover, CXCL5, CXCL8, and CXCL1 are also potent chemoattractants for neutrophils by activation of CXCR1 and CXCR2 of neutrophils [88]. Upregulation of CXCL1 and CXCL3 has been found to potentiate infiltration of immunosuppressive neutrophils, favoring cancer cells escaping from immune surveillance [31]. 

It is well known that CXC chemokines are potent drivers of leukocytes for tumor-promoting inflammation. Therefore, CXC chemokines are important regulators in the communication between tumor and leukocytes and cancer-associated inflammation. For example, neutrophil-derived IL-8 and tumor-derived CXCL1 are involved in the interaction of tumor and the LPS-stimulated neutrophils, resulting in tumor cell extravasation and distant metastasis [95]. 

However, the inflammatory microenvironment of different tumors is regulated by different chemokines. In addition, different chemokines chemoattract different inflammatory cells. Therefore, CXC chemokines may be used as a therapeutical target for disrupting tumor-promoting inflammatory milieu.

**Table 3 cancers-15-00167-t003:** Specificity of the stimulus for CXC chemokine expression and synthesis.

CXC Chemokine	Stimulus *				References
	LPS	TNF	IL-1	IFN-γ	
CXCL1/GRO-α	+	+	+	-	[96]
CXCL2/GRO-β	+	+	+	ND	[49,97,98]
CXCL3/GRO-γ	+	+	+	ND	[84,99,100]
CXCL4/PF4	+	ND	ND	ND	[101]
CXCL5/ENA-78	+	+	+	-	[96]
CXCL6/GCP-2	+	+	+	-	[102]
CXCL7/NAP-2	+	+	+	+	[84,103]
CXCL8/IL-8	+	+	+	-	[96,102]
CXCL9/MIG	+	-	-	+	[96,100]
CXCL10/IP-10	+	+	+	+	[96]
CXCL11/I-TAC	+	+	+	+	[104,105]

* can be induced (+), cannot be induced (-), ND: not determined.

## 5. CXC Chemokines and Tumor Metastasis

Metastasis is one of the major impediments for cancer treatment and is a major factor for cancer patient death. Tumor metastases begin with basement membrane degradation by matrix metalloproteinases (MMPs), such as MMP2 and MMP9, followed by tumor extravasation and dissemination, and finally colonization in new lesions. It has been reported that CXC chemokines actively participate in distant colonization of cancer cells in tumor metastasis. However, the exact mechanism by which CXC chemokines regulate tumor metastasis remains to be determined. 

Tumor-associated inflammation and neutrophil infiltration are critical for the establishment of tumor metastasis. It has been shown that CXC chemokines play crucial roles in neutrophil activation and recruitment. When tumor cells are co-cultured with neutrophils, the expression of many pro-metastatic genes, including chemokine receptors (CXCR4 and CXCR7), MMPs (MMP12 and MMP13), and growth factors (IL-6 and TGF-β), is significantly elevated [106]. It was observed that the release of CXCL6 contributes to recruiting neutrophils loaded with proteases that promote tumor invasion and metastasis [6].

It is well known that angiogenesis is essential for tumor metastasis by providing avenues for cancer cell migration. CXCL8 is a potent mediator for tumor angiogenesis. Studies have confirmed that CXCL8 enhances vessel density and facilitates distant metastasis in melanoma. In vivo studies, it was found that CXCL8 administration increased metastasis in a melanoma mouse model [83]. It has been demonstrated that the effect of CXCL8 on facilitating tumor metastasis is mediated by PI3K, AKT, and ERK signaling pathways [32].

Epithelial–mesenchymal transition (EMT) is an important process for tumor metastasis. It was observed that CXCL1 was significantly increased in the metastatic lesions which results in an elevation of mesenchymal markers expression including β-catenin, vimentin, and N-cadherin and a decrease in epithelial marker E-cadherin expression [8,34]. Similarly, the addition of recombinant CXCL5 to papillary thyroid carcinoma cells downregulated epithelial cell markers and upregulated mesenchymal markers through activating β-catenin signaling [35].

Extracellular matrix degradation is a direct cause for tumor dissemination and invasion. Extracellular matrix degradation is usually catalyzed by several proteases, such as MMPs and serine protease plasmin [107]. Thus, MMPs are crucial for cancer cells extravasation. It has been shown that CXCL1 is significantly upregulated in estrogen receptor-negative breast cancer and promotes cell migration and invasion through the ERK/MMP2/MMP9 pathway [70]. Studies have also shown that knocking down CXCL1/CXCL2 by short hairpin RNA caused a reduction in metastasis from mammary tumors to the lungs [49]. Similarly, CXCL12/CXCR4 activation also increased MMP2/MMP9 and urokinase-type plasminogen activator expression by the Wnt/β-catenin pathway, enhancing cancer metastatic tendency [46]. In penile cancer, enforced expression of CXCL13 stimulates tumor metastasis by inducing the expression of metastasis-related MMP2/MMP9 through STAT3 and ERK1/2 signaling pathways [47]. 

Additionally, cell adhesion molecules also actively participate in the distant metastasis of tumor cells. Elevated expression of vascular adhesion molecule-1 (VCAM-1) was proven to potentiate CXCL13-mediated metastasis [30]. Meanwhile, the epithelial cell adhesion molecule (EpCAM), a cell surface molecule, is found to be overexpressed in the majority of human epithelial cancers and improve tumor invasion. Studies have indicated that IL-8 exerts a critical role in EpCAM-dependent breast cancer invasion [108]. When IL-8 is neutralized, EpCAM-mediated invasion will be abrogated, while upregulation of IL-8 leads to enhanced invasion. In addition, it has been shown that type III collagen is required for the metastasis of pancreatic cancer, while CXCL3 is the factor in pancreatic cancer that enhances collagen III expression by promoting cancer-associated fibroblasts transformation [9].

Furthermore, the premetastatic niche (PMN) is an important tissue microenvironment for tumor metastasis. It has been found that CXCL1 is active in PMN formation. CXCL1 attracts hematopoietic stem/progenitor cells (HSPCs), facilitates the differentiation from HSPCs to myeloid-derived suppressor cells (MDSCs), promotes PMN formation [109], and enables tumor cells to escape the anti-tumor immune defense [94]. MDSCs are involved in CXCL1/2-induced metastasis through the paracrine pathway. Breast cancer cell-derived CXCL1/2 recruits MDSCs to produce S100A8/9, thus augmenting the metastatic tendency of breast cancer cells [49].

In addition, the chemokine–cholesterol signaling axis also plays roles in tumor metastasis. Studies have shown that the TNBC cells-derived chemokines CXCL1/CXCL2/CXCL8 stimulate cholesterol synthesis, facilitate angiogenesis, and enhance TNBC cells lung metastases [37].

In conclusion, CXC chemokines regulate cytokine secretion and gene expression and produce and maintain a pro-metastatic microenvironment.

## 6. Application of CXC Chemokines for Cancer Diagnosis, Prognosis, and Therapy

At present, many cancers still lack reliable biomarkers for diagnosis and prognosis, which becomes a major hurdle for superior cancer treatment. Therefore, identification of new cancer biomarkers is an urgent task for cancer therapy. CXC chemokines are secretory proteins primarily in the blood, which makes CXC chemokines detectable with simple, fast, and no-invasive technical approaches. As a significant amount of CXC chemokines presents in the blood, serum samples have been widely used to detect CXC chemokine levels by ELISA [110]. CXC chemokines in tissue samples have been detected by immunohistochemical staining techniques [80,102] and microarray analysis [111]. However, CXC chemokines may also exist in other samples, such as plasma [112], platelets [112], urine [113], pleural effusion, and ascitic fluid. Detection of CXC chemokines in these samples has not been reported and needs to be developed. Tumor cells have complex chemokine signaling networks that regulate cell growth, survival, angiogenesis, and metastasis. Thus, CXC chemokines are ideal biomarkers used for diagnosis and prognosis in clinic. However, combinations of CXC chemokine levels and other clinical indexes may provide better guidance for cancer treatment. The value of urine CXCL1 concentration normalized by urine creatinine (CXCL1/Cre) was demonstrated to have great advantage in predicting post-transurethral resection recurrence of bladder cancer patients. The study showed that patients with higher CXCL1/Cre had a higher risk of developing intravesical recurrence after transurethral resection [114]. Moreover, the detrimental effect of CXCL12/CXCR4 is found to depend on the density of tumor-infiltrating CD8-positive T lymphocytes. Thus, the expression of CXCL12/CXCR4 combined with the density of CD8-positive T-lymphocytes may enhance the prognostic significance of CXCL12/CXCR4. It was observed that high expression of CXCL12/CXCR4 and low density of CD8-positive T-lymphocytes are more likely to cause shorter overall survival in thyroid cancer [115].

Furthermore, dysregulated expression of CXC chemokines has been observed to be associated with the clinicopathological features and clinical outcomes of cancer patients. Overexpression of tumor-promoting CXC chemokines in cancer patients is accompanied by aggressive phenotypes (large tumor size, distant metastasis, and advanced stage) and inferior prognosis. More recent work has suggested that CXCL5 is a biomarker for poor prognosis in NSCLC patients [33,86]. In particular, patients with high expression levels of CXCL5 are more likely to have lymph node metastasis, a higher tumor stage, and shorter overall survival (OS) and progression-free survival (PFS). In addition, CXCL1, CXCL7, and CXCL8 are also upregulated in NSCLC which were positively associated with worse OS. A study observed that the levels of CXCL1, CXCL2, CXCL5, CXCL7, and CXCL8 were higher in lung cancer compared to multiple other cancer types (breast, colorectal, esophageal, head and neck, and liver cancer) [116]. Similarly, CXCL4, CXCL8, CXCL9, CXCL10, and CXCL11 are increased in early-stage NSCLC patients [117]. Additionally, in ovarian cancer (OC) patients, circulating CXCL1, CCL4, and CCL20 are elevated in serum specimens, which are associated with shorter recurrence-free survival (RFS) and OS. In addition, circulating CXCL9 and CXCL10 are also highly elevated in OC serum which is correlated with shorter RFS [118]. In TNBC, enhanced release of CXCL1/CXCL2/CXCL8 is associated with lung metastasis and neovascularization [37]. In cervical cancer, upregulation of CXCL1/CXCL2 is associated with malignant tumor phenotypes (larger tumor size, lymph node (LYN) metastasis, and shorter patient survival) [119]. In gastric cancer, enforced expression of CXCL13 is positively correlated with cancer progression and poor survival [120]. Immunohistochemistry analysis of tumor samples from gastric cancer patients has shown that high expression of CXCL1 is correlated with inferior survival and metastatic tendencies [36]. In pancreatic endocrine tumors, CXCL12 is essential for tumor progression. High expression of CXCL12 is associated with aggressive features, such as large tumor size, vascular invasion, hematogenous metastasis, large solid nests, and poor survival [121] while in clear cell renal cell carcinoma (ccRCC), patients with high levels of CXCL7 had longer PFS and OS compared to patients in which these levels were low [122]. 

In addition, the level of CXC chemokines in tumor tissues may be used as predictive indexes for cancer prognosis. A study investigating the relationship between the levels of several CXC chemokines in tumor tissues and gastric cancer relapse has shown that the levels of CXCL1, CXCL2, CXCL5, CXCL8, CXCL11, and CXCL13 in gastric cancer patients without recurrence after treatment were significantly lower than those before treatment, and there were no changes in most of the CXC chemokines in patients with recurrence [24]. Moreover, CXCL13 expression stratified gastric cancer patients in T2–4 stage. Low expression of CXCL13 in T2–4 patients predicted a better chemotherapy response [120]. Plasma levels of IL-8, CXCL9, and CXCL10 were greatly increased in patients with gastric cancer, while being decreased after surgery [111]. Thus, the level of CXC chemokines in tumor tissues may be used for predicting treatment efficacy and monitoring cancer prognosis.

Recently, the therapeutic value of CXC chemokines for multiple cancers has been explored. The expression level of some CXC chemokines would change in the process of drug administration. It was observed that activation of CXCR4 by CXCL12 enforced tumor resistance to therapies by decreasing apoptotic signaling [40]. CXCL1, CXCL2, CXCL8, and CXCL12 were observed to diminish tumor response to chemotherapy, especially in breast cancers [49,123,124]. Under chemotherapy, the CXCL1/2–S100A8/9 survival signaling axis is hyper-activated, which decreases the sensitivity of cancer cells to therapeutic medicine [49]. In addition, in various cancers, high release of CXCL8 was associated with the poor effects of drugs (e.g., oxaliplatin, 5-fluorouracil, paclitaxel, and camptothecin) [125,126,127,128,129,130]. Therefore, inhibition of these chemokines may sensitize tumor cells to chemotherapies. In addition, CXC chemokines may be used as indicators of drug efficacy. As some of the CXCLs/CXCRs signaling pathways have tumor-promoting effects, suppression of these signaling pathways provides effective approaches for cancer targeted therapy. It has been proposed that activation of the CXCL8–CXCR1/2 pathways confers resistance to chemotherapies in breast cancer, prostate cancer, and colorectal carcinoma [32]. Thus, neutralizing antibodies to CXCL8 or antagonists towards CXCR1/2 to disrupt CXCL8–CXCR1/2 interaction may improve the efficacy of current cancer treatment. In addition, some CXC chemokines have anti-tumor activities. For example, CXCL10 was found to stimulate anti-tumor immunity and thereby suppress myeloma [55]. More recently, CXCL10-Ig was used to treat myeloma and it performed well in triggering immune systems to defend tumor cells and significantly decreasing tumor growth [131]. Recombinant platelet factor-4 variant chemokine CXCL4L1 was shown to significantly suppress tumor growth and metastasis by inhibiting angiogenesis [132].

CXC chemokines are differentially expressed in different types of cancer (Table 4). It has been found that tumors may have unique chemokine profiles or molecular fingerprints that are specific to cancer types [117]. For example, in colorectal cancer tissue, CXCL1, CXCL2, CXCL3, CXCL5, CXCL8, and CXCL11 were highly expressed, while CXCL12, CXCL13, and CXCL14 were little expressed [133]. Furthermore, the production of CXC chemokines is differentially regulated in terms of tumor stage and localization. CXCL4 is a known chemokine to have an anti-tumor effect [21]. However, in the early stages of some tumors (e.g., liposarcoma, osteosarcoma, and mammary adenocarcinoma), CXCL4 is upregulated in platelets and facilitates tumor progression [112]. Moreover, the level of CXCL4 in platelets remains elevated but there is no increase in plasma during the early development of tumors [112]. Unlike CXCL4, the level of CXCL5 is greatly enhanced in serum in the late stages of gastric cancer [110]. Similarly, a high level of CXCL1 is also seen in late-stage and node-positive gastric cancer [134]. In addition, the level of IL-8 in gastric tumor tissues is higher in advanced gastric cancer than in early gastric cancer [135]. IL-8 expression was also increased in colorectal cancer patients with high-stage, deep local invasion and lymph node metastasis [128]. In advanced esophageal adenocarcinoma, the level of IL-8 is higher than that in the early stage [136]. In hepatocellular carcinoma, elevated serum levels of CXCL13 are positively associated with large tumor size and the late stage of the cancer [137]. Therefore, the level of CXC chemokines in different types of cancer and different stages of tumors may serve as a clinical parameter for cancer targeted therapy and individualized treatment. 

Taken together, manipulating CXC chemokine levels is an attractive approach for cancer treatment. Expression profiles of CXC chemokines or circulating levels of chemokines in the blood are useful tools for cancer diagnosis and prognosis. As CXC chemokines have dual effects in carcinogenesis, such as the downregulation of the tumor-promoting CXC chemokines with RNA interference techniques or chemical inhibitors or neutralizing antibodies, and the upregulation of the tumor-suppressing CXC chemokines by exogenous expression or the addition of recombinant proteins, they may provide effective approaches for cancer therapy.

## 7. Conclusions

CXC chemokines play important roles in tumor initiation and progression. CXC chemokines, especially ELR+ CXCLs, promote tumor neovascularization. Interaction of CXC chemokines with tumor tissues produces a tumor metastatic microenvironment. Cancer patients with tumors constitutively expressing CXC chemokines usually have a poor prognosis. Although it has been observed that CXC chemokines have significant effects on tumor proliferation, angiogenesis, migration, invasion, and metastasis, the regulatory mechanisms underlying the effects of CXC chemokine on cancer onset and development remain incompletely determined.

Regulation of angiogenesis is one of the major effect of CXC chemokines on tumor progression [84]. CXC chemokines recruit endothelial cells to tumor tissues and enhance their proliferation, thus promoting angiogenesis [22,85]. At the same time, CXC chemokines elevate the expression of pro-angiogenic genes, such as VEGF [88], FGF, and MMPs [85], in tumor tissues to facilitate tumor growth. 

CXC chemokines are pro-inflammatory mediators and are chemotactic for leukocytes (e.g., neutrophils, monocytes, macrophages, lymphocytes, and eosinophils). These cells are critical for inflammatory and immune responses [3]. CXC chemokine-mediated inflammation has been demonstrated to play important roles in tumor development, progression, metastasis, and immunosuppression, as well as treatment resistance [89]. 

In addition to the roles of CXC chemokines in facilitating angiogenesis and modulating inflammatory and immune responses, they also regulate tumor metastasis. Many studies have shown that CXC chemokines attract neutrophils to tumor tissues, promote neovascularization, enhance the expression of pro-metastatic genes (e.g., SPARC, CXCR4, PTGS2/COX2, ANGPT, ALDH, EREG, and EFEMP) [69], and thereby produce a pro-metastatic microenvironment [106]. 

It has been proposed that inducing angiogenesis, producing a tumor-promoting inflammatory response, and activating metastasis are hallmarks of cancer progression [25,146]. CXC chemokines are the factors driving these tumor processes. Therefore, CXC chemokines have great potential to be used in future cancer therapies.

## Figures and Tables

**Figure 1 cancers-15-00167-f001:**
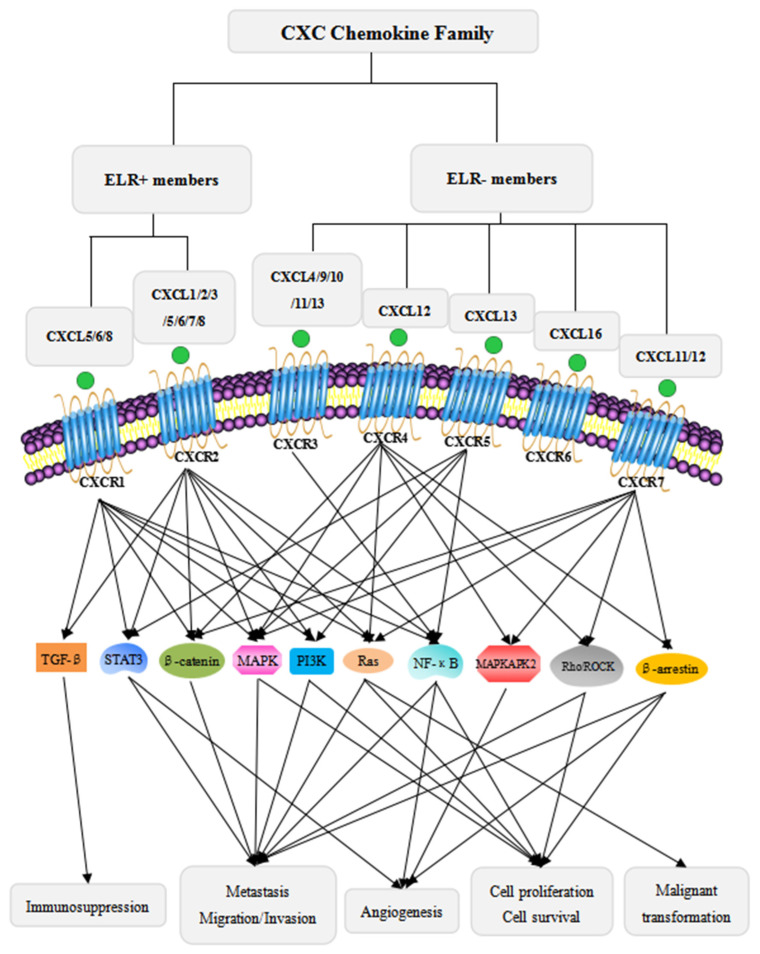
Classification of the CXC chemokine family and the major signaling pathways of CXC chemokines in cancer progression. The CXC chemokine family is divided into two subtypes: ELR+ members and ELR- members [21,22,23]. ELR+ members bind to CXCR1/2 [21,26], and ELR- members bind to CXCR3/4/5/6/7. ELR+ members promote tumor progression through multiple signaling pathways. The ELR+ CXCLs/CXCRs axes specifically activate TGF-β signaling with immunosuppressive activity [31]. Moreover, the ELR+ CXCLs/CXCRs signaling axes activate MAPK [14,32,33,34], PI3K/AKT [11,17,33], β-catenin [35], STAT3 [7,36], NF-κB [8,27,37], and Ras [20] signaling pathways to mediate tumor cells proliferation, metastasis, angiogenesis, and malignant transformation. Although ELR- members mainly have anti-tumor activity, some ELR- members have dual roles in cancer progression. The ELR- CXCLs/CXCRs axes activate MAPKAPK2 signaling to enhance angiogenic activity [38], Rho/ROCK signaling to increase metastatic tendencies and immortal proliferation [39,40], and β-arrestin signaling to promote metastasis, immortal pro iferation and angiogenesis [41]. In addition, ELR- members, similar to ELR+ members, also activate MAPK [40,42,43,44], PI3K/AKT [40,42,44,45], β-catenin [46], STAT3 [47], NF-κB [30], and Ras [43] signaling pathways.

**Table 1 cancers-15-00167-t001:** CXC chemokine effects in physiological and pathological condition.

CXC Chemokine	Alternative Name	Receptor	Effects	References
CXCL1	Melanoma growth -stimulating activity (MGSA)-α/Growth-related oncogene (GRO)-α	CXCR2	(1) Promote tumor proliferation, angiogenesis, inflammation, and migration/invasion; (2) Immunosuppression	[8,13,20,31,48]
CXCL2	MGSA β/GRO-β/ Macrophage inflammatory protein-2 (MIP-2)	CXCR2	(1) Promote tumor progression, angiogenesis, and metastasis	[20,21,22,49]
CXCL3	MGSA γ/GRO-γ	CXCR2	(1) Promote tumor growth, angiogenesis, and metastasis; (2) Immunosuppression	[9,20,21,22,31]
CXCL4	Platelet factor 4 (PF4)	CXCR3	(1) Inhibit angiogenesis; (2) Promote angiogenesis in some cancers (colorectal cancer)	[21,22,27]
CXCL5	Epithelial-derived neutrophil-activating factor-78 (ENA-78)	CXCR1 CXCR2	(1) Promote tumorigenesis, angiogenesis, inflammation, and metastasis	[20,33,35,50]
CXCL6	Granulocyte chemotactic protein-2 (GCP-2)	CXCR1 CXCR2	(1) Promote tumor growth and angiogenesis; (2) Antibacterial activity	[20,51,52]
CXCL7	Neutrophil-activating polypeptide-2 (NAP-2)	CXCR2	(1) Promote angiogenesis	[20,21,22]
CXCL8	Interleukin-8 (IL-8)	CXCR1 CXCR2	(1) Promote tumor growth, angiogenesis, inflammation, and metastasis; (2) Immunosuppression; (3) Chemoresistance	[20,32,50]
CXCL9	Monokine induced by interferon-γ (MIG)	CXCR3	(1) Inhibit angiogenesis	[21,22,53,54]
CXCL10	Interferon-inducible protein 10 (IP-10)	CXCR3	(1) Inhibit angiogenesis; (2) Enhance anti-tumor immunity	[21,22,53,54,55]
CXCL11	Interferon-inducible T cell α chemoattractant (I-TAC)	CXCR3	(1) Inhibit angiogenesis	[21,22,41,54]
CXCL12	Stromal cell-derived factor-1 (SDF-1)	CXCR4 CXCR7	(1) Promote tumor growth, angiogenesis and metastasis and lead to poor prognosis; (2) Involved in hematopoietic stem cell survival, proliferation, and homing; (3) Repair damaged tissue	[28,38,40,41]
CXCL13	B-lymphocyte chemoattractant (BLC)	CXCR3 CXCR5	(1) Promote tumor proliferation, migration, metastasis, and recurrence	[47,56,57]

**Table 2 cancers-15-00167-t002:** Corresponding signaling pathways and tumor effects of CXC chemokines.

CXC Chemokines	Signaling Pathways	Tumor Effects	References
CXCL1	(1) P38/ERK/MAPK pathway; (2) VEGF/STAT3 pathway; (3) Integrin β1/FAK/AKT signaling pathway; (4) NF-κB pathway; (5) Ras signaling pathway	(1) Promote proliferation of ovarian cancer; induce breast cancer cells migration/invasion; facilitate uterine cervix cancer malignant processes; (2) Promote gastric cancer cells angiogenesis; (3) Promote gastric cancer cells lymph node metastasis; (4) Promote breast cancer cells and gastric cancer cells’ migration/ invasion; (5) Promote melanocytes malignant transformation	[8,11,12,14,20,34,36,69,70]
CXCL5	(1) PI3K/AKT and ERK/MAPK signaling pathways; (2) β-catenin pathway; (3) NF-κB pathway	(1) Promote proliferation and migration of non-small cell lung cancer cells; (2) Promote the migration/invasion and EMT of papillary thyroid carcinoma cells; (3) Promote growth and progression of prostate cancer	[33,35,71]
CXCL8	PI3K/AKT and ERK/MAPK signaling pathways	(1) Support survival and proliferation of acutemyeloid leukemia cells; enhance tumor (melanoma) growth and metastasis	[17,32,60]
CXCL1/3	TGF-β signaling pathway	(1) Drive immunosuppressive neutrophils infiltration	[31]
CXCL1/5	CXCR2/STAT3 pathway	(1) Induce EMT and metastasis of gastric cancer cells	[7]
CXCL1/2/4	NF-κB pathway	(1) Promote tumor growth and angiogenesis in colorectal cancer cells	[27]
CXCL1/2/8	NF-κB pathway	(1) Promote TNBC lung metastasis	[37]
CXCL12	(1) MAPKAP Kinase-2 signaling; (2) ERK and AKT pathway; (3) Rho/ROCK pathway; (4) Ras signaling pathway; (5) Wnt/β-catenin pathway	(1) Promote angiogenesis in colon tumors; (2) Promote proliferation and survival of breast cancer cells; (3) Facilitate cancer growth and EMT; (4) Increase cell proliferation in pancreatic cancer; (5) Promote metastasis in colorectal cancer	[38,39,40,42,43,46]
CXCL11/12	β-arrestin signaling	(1) Promote cancer cell growth, angiogenesis, and metastasis	[41]
CXCL13	NF-κB pathway, PI3K/AKT pathway, ERK/MAPK pathway, STAT3 pathway	(1) Promote cancer cell growth and metastasis	[30,44,45,47]

**Table 4 cancers-15-00167-t004:** Different cancer types and their corresponding CXC chemokines.

Cancer Type	CXC Chemokine	Effects	References
Lung cancer	CXCL1/2/5/6/7 CXCL4/8/9/10//11(early-stage)	Tumor promotion	[81,116,117,138]
Gastric cancer (GC)	CXCL1/13 CXCL5/8 (late-stage)	Tumor promotion. CXCL13 could be used to stratify GC patients in the T2–4 stage, and low expression of it predicts better therapy response	[110,120,134,135]
Colorectal cancer (CRC)	CXCL1/2/3/5/8/11 (up) CXCL12/14 (down)	Tumor promotion	[133]
Esophageal adenocarcinoma	CXCL8 (late-stage)	Tumor promotion	[136]
Oral squamous carcinoma	CXCL1	Tumor promotion	[13]
Pancreatic carcinoma	CXCL3/8/12	Tumor promotion	[9,85,121]
Hepatocellular carcinoma	CXCL1/3/5 CXCL13 (late-stage)	Tumor promotion	[10,137]
Triple-negative breast cancer (TNBC)	CXCL1/2/8	Tumor promotion	[37]
Cervical cancer	CXCL1/2	Tumor promotion	[119]
Ovarian cancer	CXCL1/9	Tumor promotion	[118]
Prostate cancer	CXCL8/13	Tumor promotion	[32,44]
Bladder cancer	CXCL1/5/8/12	Tumor promotion	[113,114,139,140]
Clear cell renal cell carcinoma	CXCL7 (down)	Tumor promotion	[138]
Melanoma	CXCL1/8	Tumor promotion	[20,141]
Head and neck squamous cell carcinoma (HNSCC)	CXCL5/8	Tumor promotion	[142,143]
Thyroid cancer	CXCL8/12 CXCL10 (down)	Tumor promotion	[115,144,145]
Liposarcoma, Osteosarcoma	CXCL4 (early-stage)	Tumor promotion	[112]
B-cell acute lymphoblastic leukemia	CXCL13	Tumor promotion	[57]

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
