# Peer review of "The Role of CXC Chemokines in Cancer Progression"

_cancers, 2022, doi:10.3390/cancers15010167_

Round 1

Reviewer 1 Report

Comments: The work is an interesting manuscript,  However few conceptual issues should to be addressed ,

1.     Authors didn’t discuss if there were any symptoms  related to  increase level of CXC in patients.

2.     The relation between other inflammatory chemokines and CXC was not outlined if the patient was suffered from cancer or not. and how can clinical physician   distinguish between them?

3.     The methods that can be used to identify CXC  in clinic should have been summarized to provide very clear aspects for reader

4.     In section "Application of CXC Chemokines for Cancer Diagnosis, Prognosis and Therapy " can authors provide  table to summarize each type of cancer and its  CXC corresponding  type

5.     Mechanism by which anti CXC can inhibit and block CXC was not studied. Authors should to provide paragraph for how antagonist can block/prevent signalling pathway of CXC

Author Response

Point 1: Authors didn’t discuss if there were any symptoms related to increase level of CXC in patients.

Response 1: Upregulation of CXC chemokines is closely associated with poor survival of cancer patients (discussed in the section 6). Recent studies mainly focused on association of expression of CXC chemokines with the clinicopathological features and prognosis. High expression of CXC chemokines is positively associated with large tumor size, increased metastasis, tumor recurrence and short survival. However, the clinical symptoms related to enhanced expression of CXC chemokines were barely discussed.

Point 2: The relation between other inflammatory chemokines and CXC was not outlined if the patient was suffered from cancer or not. and how can clinical physician distinguish between them?

Response 2: Very interesting point. The relation between other inflammatory chemokines and CXC chemokines in tumor inflammation is a big topic and beyond the scope of this review. However, we have added a few sentences in “Introduction” to briefly discuss the relation between the chemokines in tumors. Currently, very little is known about the cancer symptoms specifically associated with these chemokines. In addition, CXC chemokines should be used in combination with other indexes as biomarkers for predicting cancer prognosis and evaluating therapeutic efficacy.

Point 3: The methods that can be used to identify CXC in clinic should have been summarized to provide very clear aspects for reader.

Response 3: Detection methods for CXC chemokines have been added into the section "Application of CXC Chemokines for Cancer Diagnosis, Prognosis and Therapy ". As CXC chemokines are secretory proteins, they are mainly detected by ELISA in serum samples.

Point 4: In section "Application of CXC Chemokines for Cancer Diagnosis, Prognosis and Therapy " can authors provide  table to summarize each type of cancer and its  CXC corresponding type.

Response 4: Thanks for your suggestion. The table has been added to this section (see Table 4).

Point 5: Mechanism by which anti CXC can inhibit and block CXC was not studied. Authors should to provide paragraph for how antagonist can block/prevent signalling pathway of CXC.

Response 5: Thanks for your suggestion. I have added this content in the section “CXC Chemokine Family and the CXCL/CXCR Signaling Axes”.

Reviewer 2 Report

Leukocyte migration to inflammatory regions and secondary lymphoid organs is mediated by chemokines, also known as chemotactic cytokines, and their receptors. They are crucial for tumor genesis, development, and advancement in addition to their roles in the immune system. There are four subgroups of chemokines: CXC, CC, CX3C, and C chemokine ligands. This review is focusing on the role of CXC chemokines in tumor initiation and development and possible applications of CXC chemokines for cancer treatment. It is an important research topic. Therefore, the authors have made a detailed literature analysis and presented it in a reasonable and organized manner. However, I have some suggestions and corrections to the article that are appended below.

Comments

Point 1: Graphical abstract is needed to show the overview of the manuscript.

Point 2: Abstract is too concise; there is a need to add a few lines about future directions.

Point 3: Introduction: The information described in this section is appropriate and exhaustive to introduce the following sections. To increase the importance of this review, add the novelty of this review article to differentiate it from the other available articles.

Point 4: As the study is based on reports already available in the literature, it does not present novel data in its present form. Instead, it contains an extensive presentation of topics. Various reviews are available as https://doi.org/10.1016/j.canlet.2008.04.050, https://doi.org/10.1074/jbc.RA119.010018, https://doi.org/10.1016/j.bbcan.2011.10.008, https://doi.org/10.1016/S1050-1738(97)00128-X.

Point 5: In the tabular form, express the "Specificity of the stimulus for CXC chemokine expression and synthesis" in the context of LPS            TNF     IL-1     IFN-γ.

Point 6:  Another problem is the need for concision; it is too descriptive, monotonous and has much redundant information.

Point 8: This review does not discuss the relationship between Chemokines and their role in metastasis and drug resistance.

Author Response

Point 1: Graphical abstract is needed to show the overview of the manuscript.

Response 1: Thanks for your suggestion. The graphical abstract has been attached with the review.

Point 2: Abstract is too concise; there is a need to add a few lines about future directions.

Response 2: Thanks for your suggestion. we have revised the abstract based on the comment.

Point 3: Introduction: The information described in this section is appropriate and exhaustive to introduce the following sections. To increase the importance of this review, add the novelty of this review article to differentiate it from the other available articles.

Response 3: Thanks for your suggestion. Based on your suggestion, we have polished the introduction.

Point 4: As the study is based on reports already available in the literature, it does not present novel data in its present form. Instead, it contains an extensive presentation of topics. Various reviews are available as https://doi.org/10.1016/j.canlet.2008.04.050, https://doi.org/10.1074/jbc.RA119.010018, https://doi.org/10.1016/j.bbcan.2011.10.008, https://doi.org/10.1016/S1050-1738(97)00128-X.

Response 4: Thanks for your suggestion. I have read the articles you recommended and other related articles and revised the review accordingly.

Point 5: In the tabular form, express the "Specificity of the stimulus for CXC chemokine expression and synthesis" in the context of LPS, TNF, IL-1, IFN-γ.

Response 5: Thanks for your suggestion. A table about the specificity of the stimulus on CXC chemokines has been added to the section “CXC Chemokines and Tumor-Associated Inflammation”.

Point 6: Another problem is the need for concision; it is too descriptive, monotonous and has much redundant information.

Response 6: Thanks for your suggestion. The review has been revised accordingly.

Point 7: This review does not discuss the relationship between Chemokines and their role in metastasis and drug resistance.

Response 7: I have now added a portion discussing association of CXC chemokines with tumor metastasis in the section 5 “CXC Chemokines and Tumor Metastasis”. Association of CXC chemokines with therapeutic resistance has been discussed in the section 6 "Application of CXC Chemokines for Cancer Diagnosis, Prognosis and Therapy " .

Reviewer 3 Report

In this article, the authors have described the  roles of CXC chemokines in tumor initiation and development and possible applications of CXC chemokines for cancer treatment. The article has been presented with many subheads including  the CXC Chemokine Family and the CXCL/CXCR Signaling Axes,  CXC Chemokines and Tumor Angiogenesis,  CXC Chemokines and Tumor-Associated Inflammation,  Application of CXC Chemokines for Cancer Diagnosis, Prognosis and Therapy and Conclusions. The authors have presented one illustration and two tables. The bibliography is appropriate, and the citations are quite recent.  The article is quite well and recent and beneficial to readers of the field.

Comment :

The authors should present an illustration showing the potential application of CXC chemokines for cancer-targeted therapy and  personalized treatment.

Author Response

Point 1: The authors should present an illustration showing the potential application of CXC chemokines for cancer-targeted therapy and personalized treatment.

Response 1: Thanks for your suggestion. Currently, impairing the CXCL/CXCR signaling by antagonists and neutralizing antibodies is the major approach for the CXC chemokine-related cancer targeted therapy. Other applications utilizing CXC chemokine signaling for cancer treatment are still under development. As it is too straightforward for the current CXC chemokine therapeutic strategy, we did not present an illustration. Nevertheless, we have add a table (Table 4) to summarize cancer type and the corresponding CXC chemokines as a reference for the CXC chemokine-related cancer therapy in the section "Application of CXC Chemokines for Cancer Diagnosis, Prognosis and Therapy ".

Round 2

Reviewer 1 Report

Good response and very interesting review 

Reviewer 2 Report

·       Most of the suggestions have been incorporated by the authors in the revised manuscript. Therefore, no issue with considering it for publication.